# Hyperglycemia and bladder cancer prognosis in a Finnish population-based cohort

Lauri Vuoristo[1], Antti Pöyhönen [2], Silja M. Vuorlaakso[1], Andres Kotsar[3],
Teuvo L.J. Tammela[1,4], Teemu J. Murtola[1,4]*

1 Faculty of Medicine and Health Technology, University of Tampere, Tampere, Finland, 2 The Finnish Defence Forces, Centre for Military Medicine, Riihimäki, Finland, 3 Department of Urology, Tartu University Hospital, Tartu, Estonia, 4 Department of Urology, TAYS Cancer Center, Tampere, Finland

* teemu.murtola@tuni.fi

## Abstract

### Background

Diabetes might increase risk of bladder cancer (BCa) incidence and bladder cancer specific death.

### Objective

To assess the potential association between diabetes and prognosis of BCa.

### Design, setting and participants

Register-based cohort study included 14,638 participants with BCa diagnosed in Finland during 1995–2012. Three databases were used to obtain information on BCa, diabetes and comorbidities. Hyperglycemia was determined by blood glucose or HbA1c. The role of antidiabetic medication use was also assessed.

### Outcome measurements and statistical analysis

Associations between diabetes and BCa-specific and overall mortality were evaluated using multivariable Cox regression models.

### Results and limitation

During median follow-up of 4.3 years after BCa diagnosis 3,582 (24.5%) participants died of BCa. Diabetic post-diagnostic blood glucose level was associated with worse BCa-specific HR 1.58 (1.10–2.28) and overall survival HR 1.50 (1.20–1.89). The risk increase was slightly lower among hyperglycemic DM medication users (HR 1.43 (0,68–3.02) than non-users HR 1.80 (1.13–2.85). The main limitation of this study was insufficient information on tobacco smoking.

**Data availability statement:** The study utilized routinely collected data from national health-care registries, with approvals and consents obtained from the relevant registry keepers, including the Finnish Institute for Health and Welfare (THL) and the Social Insurance Institution of Finland (SII). The datasets used in this study are not publicly available, as they contain potentially identifiable personal-level data and are owned by the respective national registry keepers. Access to the data requires permission from the Finnish national data authority, FINDATA (https://findata.fi/en/).

**Funding:** The author(s) received no specific funding for this work.

**Competing interests:** I have read the journal's policy, and the authors of this manuscript declare the following competing interests: Teuvo LJ Tammela has received consultant fees from Astellas, Bayer, and Roche. Teemu J Murtola has received consultant fees from Novartis, Astellas, Janssen Cilag, Amgen, and Recordati; lecture fees from Ferring, Novartis, Sanofi, Bayer, Roche, Pfizer, Ipsen, Astellas, Amgen, and Janssen Cilag; and research funding from Bayer. All other authors declare no conflicts of interest. This does not alter our adherence to PLOS One policies on sharing data and materials.

**Abbreviations:** BCa, bladder cancer; TURB, transurethral resection of urinary bladder tumor; HR, hazard ratio; CI, confidence interval; FCR, Finnish cancer registry; ICD, international classification of diseases; HILMO, hospital discharge registry; COPD, chronic obstructive pulmonary disease; SII, social insurance institution; CCI, Charlson comorbidity index; VS, versus; HbA1c, glycated haemoglobins

## Conclusions

Diabetes and diabetic glycemic status were associated with increased risk of both overall and BCa-specific death.

## Patient summary

In a Finnish population-based cohort study, diabetic bladder cancer patients had worse prognosis compared to non-diabetics. This was observed in both genders.

## Introduction

Bladder cancer (BCa) is the ninth most common cancer worldwide [1]. Well-known risk factors are smoking, male gender, living in developed countries and chemical exposure [2]. The incidence is three- to fourfold higher in men than in women. Nevertheless, among women the disease is often more advanced at the time of diagnosis and might respond poorly to treatment [3].

Diabetes type 2 is an independent risk factor for many cancers, but the role on risk, prognosis and recurrence of BCa is unclear [4]. The suggested mechanisms behind the worse oncological outcomes and elevated cancer incidences in diabetics include hyperinsulinemia combined with insulin resistance and hyperglycemia [5]. These presumably support growth of cancer cells by offering energy and growth factors for rapid cell growth.

According to a meta-analysis of 36 studies, diabetes was associated with an increased risk of BCa in men [6]. A prospective population-based cohort study also found an increased risk of BCa mortality, again only in men [7]. Conversely, one study involving 1,000 patients found no association between diabetes (types 1 or 2) and the risk of BCa or mortality [8].

We conducted a population-based cohort study to investigate the associations between diabetes and BCa-specific mortality, hypothesizing that individuals with diabetes would experience worse disease-specific survival.

## Materials and methods

### Study cohort

Data was obtained from the Finnish Cancer Registry (FCR) for all newly diagnosed BCa cases in Finland 1995–2012. FCR is a nationwide database containing more than 99% of all diagnosed malignancies in Finland since 1953 [9]. BCa cases were identified based on ICD-O-3 site codes 670–680, 688 and 689. Most common tumor morphologies were transitional cell carcinoma in 12,027 (82.0%), papillary transitional cell carcinoma 1,626 (11.1%) and non-specified malignant neoplasm 533 (3.6%). Ten cases diagnosed under the age of 20 were excluded. Additionally, 182 cases were excluded in which the BCa diagnosis was made at the time of death or thereafter, and these individuals did not die of BCa. In total 14,638 BCa cases were included.

Age, gender, BCa extent at the time of diagnosis (localized versus metastatic, available for 74.0% of the cohort), diagnosis date, primary treatment (surgical, unknown or other, available for 79.7%) and cause and time of death were obtained from the FCR. The endpoint information (alive, dead or emigrated) was received from Statistics Finland up until March 2014. Deaths with ICD-10 code C67 recorded as the primary cause of death were considered as BCa deaths.

Information on endoscopic and open surgical procedures performed for BCa during 1996–2012 was obtained from the nationally comprehensive hospital discharge registry (HILMO). The registry records diagnoses and procedures performed during in- and outpatient hospital visits at every medical treatment facility in Finland [10]. Surgical procedures were identified based on Nordic Classification of Surgical Procedures codes for cystectomy and for transurethral or open resection of the BCa. We were unable to obtain HILMO data from 1995 as the coding system was different before 1996.

### Information on diabetes and comorbidities

Information on diabetes diagnosis (ICD-10 codes E10 and E11) during 1996−2012 was extracted from the HILMO. Additionally, we obtained recorded diagnoses of hypertension (I10), hypercholesterolemia (E78), obesity (E65-E68), chronic obstructive pulmonary disease (COPD, J44) and smoking (Z72.0). Diagnoses made in 1995 were recorded as ICD-9 codes: 250 participants had diagnosis for diabetes, 272 for hypercholesterolemia, 401 for hypertension and 278 for obesity.

To complement the HILMO data the cohort was linked to national prescription register maintained by the Social Insurance Institution (SII) of Finland. SII is a governmental agency funded through tax revenues providing reimbursements for the costs of physician-prescribed drug purchases as part of the Finnish National Health Care Insurance. The reimbursement covers all Finns. Antidiabetic drugs (except for acarbose), antihypertensive and cholesterol-lowering drugs are reimbursed and available solely by prescription, thus recorded by the SII prescription [11].

All purchases of antidiabetic, antihypertensive and cholesterol-lowering drugs during 1995–2012 were identified from the SII prescription register using the Anatomic Therapeutic Chemical classification codes [12]. Information from FCR, Statistic Finland, HILMO and prescription database was linked using personal identification code.

We assessed the risk of comorbidity by using Charlson comorbidity index (CCI) [13]. Diagnoses for comorbidities were obtained from HILMO and supplemented with medication data from the SII prescription database.

### Information on blood glucose and glycated haemoglobin (HbA1c) levels

Information was collected on fasting glucose and glycated haemoglobins (HbA1c) levels during 1995–2012 for a subset of study population living in the Pirkanmaa region in Finland. The information on laboratory results was obtained from the Fimlab laboratory center, main provider of laboratory services in the region. We transformed all HbA1c measurements to mmol/mol. Men were defined as normoglycaemic, or diabetic separately for each follow-up years based on local and international thresholds (<53 to ≥53 mmol/mol, and ≤7 to >7 mmol/l) for yearly mean HbA1c or median fasting glucose, respectively.

### Statistical analysis

Differences in distribution of population characteristics was compared by glycemic status with Chi-square test for categorical variables and Mann-Whitney U-test for continuous variables.

To estimate the risk associations for fatal BCa hazard ratios (HRs) and 95% confidence intervals (CIs) for BCa-specific and all-cause deaths were calculated using Cox regression. Follow-up started at BCa diagnosis and ended up on death, emigration or in March 2014, whichever came first. Cox regression was adjusted for age, gender, primary treatment of BCa, tumor extent at diagnosis, hypertension, hypercholesterolemia and CCI.

In Cox regression analyses, glycemic status before and after BCa diagnosis was analyzed as a time-fixed variable taking into account all glucose and HbA1c measurements, diagnoses and drug purchases occurring between 1995 and the

year of BCa diagnosis. Glycemic status after BCa diagnosis was analyzed as time-dependent variable, where the status was updated separately for each follow-up year. As we used time-varying covariates for the analysis, checking for validity of the proportional hazards assumption over time was not necessary. Same model adjustments were used as for analysis of pre-diagnosis DM.

The time-dependent analysis was separately made for fasting glucose and HbA1c levels. For years with no blood glucose measurements available, we used the latest available blood glucose level. Participants without any measurement of fasting glucose or HbA1c levels were not included in the analysis.

We performed a sensitivity analyses limited to 10,854 cases with available information on tumor extent at diagnosis. Cancer-specific survival and overall survival were estimated by post-diagnostic glucose level.

Statistical analyses were performed using IBM SPSS statistical software version 24.

In compliance with Finnish law on medical research (9.4.1999/488) Institutional review board approval can be waived for studies based entirely on routinely collected registry data.

## Results

### Cohort characteristics

Information on use of antidiabetic medication was available for 14,638 participants; 11,151 (76.2%) men, 3,487 (23.8%) women (Table 1). Fasting glucose levels were available for a subgroup of 624 participants; 481 non-diabetic and 143 diabetic participants. Of these, 61 (12.7%) participants with non-diabetic glucose level used DM medication; 97 (67.8%) with diabetic glucose level used DM medication.

The proportion of men was higher in the diabetic group compared to non-diabetics (79.2% vs. 75.4% in DM medication users and non-users, respectively; 79.7% vs. 73.6% in participants with diabetic glucose level and normoglycemic, respectively). The median age at diagnosis was 72–73 years in all subgroups. No clear differences in Charlson co-morbidity score or prevalence of COPD were observed by medication use or blood glucose level.

Proportion of participants diagnosed with metastatic BCa was lower in DM medication users than non-users but elevated in those with diabetic glucose level (Table 1). No marked differences were observed in proportions of participants undergoing cystectomy or chemotherapy after the diagnosis.

DM medication users had similar overall mortality than non-users, while hyperglycemic participants had higher overall mortality than normoglycemic participants (Table 1). Bladder cancer mortality was lower in DM medication users than non-users, whereas hyperglycemic participants had higher bladder cancer mortality than normoglycemic participants.

Overall, 12,221 (83.5%) of the participants underwent TURB, of whom 2,608 were diabetics. Median number of TURBs after the diagnosis or length of intervals between procedure did not differ by DM medication use or glucose level (Table 1).

### Overall and cancer-specific survival by glycemia status and antidiabetic medication use

Diabetic post-diagnostic blood glucose level was associated with worse BCa-specific survival HR 1.58 (95% CI 1.10–2.28) and overall survival HR 1.50 (1.20–1.89) compared to normoglycemia in multivariable adjusted analysis (Table 2). The risk increase for BCa death was slightly lower among hyperglycemic DM medication users HR 1.43 (0,68–3.02) than non-users HR 1.80 (1.13–2.85) but remained elevated compared to normoglycemia in both groups.

Higher post-diagnostic HbA1c levels showed a similar association with increased risk of BCa death HR 1.50 (0.88–2.56) and overall survival HR 1.43 (1.06–1.93), with the result being statistically significant only for overall survival (Table 3).

Higher pre-diagnostic blood glucose level was also associated with elevated BCa death risk HR 1.41 (0.82–2.43) and overall death risk HR 1.28 (0.87–1.91), but the difference was not statistically significant (Table 4). Similarly, pre-diagnostic HbA1c levels associated with slightly elevated BCa HR 1.21 (0.65–2.25) and overall death risk HR 1.37 (0.89–2.12), also not statistically significant results.

**Table 1. Population characteristics.**

| | Antidiabetic medicine use | | Post-diagnostic blood glucose level | |
|---|---|---|---|---|
| | **No** | **Yes** | **Normoglycemic** | **Diabetic** |
| N of participants | 11623 | 3015 | 481 | 143 |
| Gender: | | | | |
| Men; n(%) | 8762 (75.4) | 2389 (79.2)* | 354 (73.6) | 114 (79.7) |
| Men who use antidiabetic medicine; n(%) | | | 47 (13.3) | 77 (67.5) |
| Women; n(%) | 2861 (24.6) | 626 (20.8)* | 127 (26.4) | 29 (20.3) |
| Women who use antidiabetic medicine; n(%) | | | 14 (11.0) | 20 (70.0) |
| Median (IQR) age at dg | 73.0 (64.0-80.0) | 73.0 (66.0-79.0) | 72.0 (64.0-80.0) | 73.0 (64.0-80.0) |
| Median Charlson co-morbidity index (IQR) | 3.0 (2.0-4.0) | 4.0 (3.0-5.0) | 4.0 (2.0-5.0) | 4.0 (3.0-6.0) |
| Tumor extent at diagnosis: | | | | |
| Localized; n(%) | 7044 (60.6) | 1926 (63.9) | 320 (66.5) | 80 (55.9) |
| Metastatic; n(%) | 1570 (13.5) | 302 (10.0) | 62 (12.9) | 30 (21.0) |
| Unknown; n(%) | 3009 (25.9) | 787 (26.1) | 99 (20.6) | 33 (23.1) |
| Median (IQR) follow-up after dg | 4.3 (1.6-8.9) | 4.9 (2.1-9.4) | 6.9 (2.6-11.3) | 5.3 (1.4-9.3) |
| Deaths; n(%) | 6807 (58.6) | 1718 (57.0) | 275 (57.2) | 96 (67.1) |
| Overall mortality; n of deaths/1000 person years | 104 | 95 | 92 | 124 |
| Bladder cancer deaths; n(%) | 3014 (25.9) | 568 (18.8) | 104 (21.6) | 39 (27.3) |
| Bladder cancer mortality; n of deaths/1000 person years | 46 | 31 | 38 | 51 |
| COPD; n(%) % | 1337 (11.5) | 312 (10.3) | 52 (10.8) | 17 (11.9) |
| Median (IQR) number of TURBs ** | 2.0 (1.0-3.0) | 2.0 (1.0-3.0) | 2.0 (1.0-3.0) | 2.0 (1.0-3-0) |
| Median (IQR) interval between TURBs (years)*** | 0.50 (0.25-1,13) | 0.58 (0.29-1.2) | 0.76 (0.35-1.5) | 0.88 (0.38-1.42) |
| Cystectomy; n(%) | 1406 (12.1) | 254 (8.4) | 103 (21.4) | 34 (23.8) |
| Chemotherapy; n(%) | 1732 (14.9) | 453 (15.0) | 90 (18.7) | 22 (15.4) |
| Radiation therapy; n(%) | 1211 (10.4) | 242 (8.0) | 25 (5.2) | 14 (9.8) |

*P for difference compared to non-diabetics < 0.05. Calculated with Chi-square test for categorical variables and Mann-Whitney U-test for continuous variables.

**the number of TURBs available for subgroups of diabetes medication users (n = 2,608) and for non-users (n = 9,451); hyperglycemic (n = 116) and normoglycemic (n = 389).

***the interval between TURBs available for subgroups of diabetes medication users (n = 1,407) and for non-users (n = 5,028); hyperglycemic (n = 59) and normoglycemic (n = 205).

**Table 2. Cancer specific and overall mortality by blood glucose level and antidiabetic medication use after bladder cancer diagnosis.**

| | | Bladder cancer death | | | Death due to any cause | |
|---|---|---|---|---|---|---|
| | Number of BCa deaths/ participants (%) | HR (95% CI)age-adjusted | HR (95% CI)multivariable adjusted* | Number of deaths/ participants (%) | HR (95% CI)age-adjusted | HR (95% CI)multivariable adjusted |
| *Overall blood glucose level, post-diagnostic* | | | | | | |
| Normoglycemic | 104/481 (21.6) | ref | ref | 275/481 (57.2) | ref | ref |
| Diabetic | 39/143 (27.3) | 1.45 (1.01–2.08) | 1.58 (1.10–2.28) | 96/143 (67.1) | 1.50 (1.19–1.87) | 1.50 (1.20–1.89) |
| *Use of antidiabetic medication* | | | | | | |
| Normoglycemic | 12/61 (19.7) | ref | ref | 13/61 (21.3) | ref | ref |
| Diabetic | 20/97 (20.6) | 1.43 (0.68–3.00) | 1.43 (0.68–3.02) | 27/97 (27.8) | 1.48 (0.99–2.22) | 1.51 (1.01–2.28) |
| *No antidiabetic medication* | | | | | | |
| Normoglycemic | 92/420 (21.9) | ref | ref | 111/420 (26.4) | ref | ref |
| Diabetic | 19/46 (41.3) | 2.49 (1.57–3.94) | 1.80 (1.13–2.85) | 27/46 (58.7) | 2.06 (1.48–2.87) | 1.48 (1.06–2.07) |

**Table 3. Cancer specific and overall mortality by HbA1c level before and after bladder cancer diagnosis.**

| | | Bladder cancer death | | | Death due to any cause | |
|---|---|---|---|---|---|---|
| | Number of BCa deaths/ participants (%) | HR (95% CI)age-adjusted | HR (95% CI)multivariable adjusted* | Number of deaths/ participants (%) | HR (95% CI)age-adjusted | HR (95% CI)multivariable adjusted |
| *HbA1c, pre-diagnostic* | | | | | | |
| < 53 (mmol/mol) | 50/347 (14.4) | ref | ref | 168/347 (48.6) | ref | ref |
| ≥ 53 (mmol/mol) | 17/80 (21.3) | 1.33 (0.73–2.42) | 1.21 (0.65–2.25) | 54/80 (67.5) | 1.48 (0.97–2.26) | 1.37 (0.89–2.12) |
| *HbA1c, post-diagnostic* | | | | | | |
| < 53 (mmol/mol) | 50/346 (14.5) | ref | ref | 168/346 (48.6) | ref | ref |
| ≥ 53 (mmol/mol) | 17/80 (21.3) | 1.25 (0.74–2.13) | 1.50 (0.88–2.56) | 54/80 (67.5) | 1.23 (0.91–1.66) | 1.43 (1.06–1.93) |

**Table 4. Cancer specific and overall mortality by blood glucose level within one year before bladder cancer diagnosis.**

| | | Bladder cancer death | | | Death due to any cause | |
|---|---|---|---|---|---|---|
| | Number of BCa deaths/ participants (%) | HR (95% CI)age-adjusted | HR (95% CI)multivariable adjusted* | Number of deaths/ participants (%) | HR (95% CI)age-adjusted | HR (95% CI)multivariable adjusted |
| *Median fasting blood glucose level before diagnosis* | | | | | | |
| Normoglycemic | 49/165 (29.7) | ref | ref | 97/165 (58.8) | ref | ref |
| Diabetic | 20/52 (38.5) | 1.32 (0.78–2.22) | 1.41 (0.82–2.43) | 37/52 (71.2) | 1.26 (0.86–1.85) | 1.28 (0.87–1.91) |

## Subgroup analysis

In a subgroup of participants, with normoglycemic blood glucose level before diagnosis, diabetic post-diagnostic blood glucose level was even stronger predictor of overall mortality than in the main analysis as HR was 2.63 (1.47–4.70) (Table 5). Additionally, the risk increase for bladder cancer death was slightly higher than in the main analysis, HR 1.96 (0.86–4.47).

Age and gender did not modify the risk association between diabetic blood glucose level and bladder cancer-specific or overall survival.

Diabetic blood glucose level predicted overall survival similarly regardless of tumor stage diagnosis. However diabetic blood glucose level predicted increased risk of bladder cancer death only in participants with advanced disease at diagnosis as HR was 2.89 (1.33–3.93), but not in participants with localized disease; HR 0.85 (0.42–1.74).

**Table 5. Cancer specific and overall mortality by post-diagnostic blood glucose level stratified by subgroups.**

| | | Bladder cancer death | | Death due to any cause |
|---|---|---|---|---|
| | Number of BCa deaths/participants (%) | HR (95% CI)multivariable adjusted* | Number of deaths/participants (%) | HR (95% CI)multivariable adjusted |
| *Overall blood glucose level, post-diagnostic.* | | | | |
| Normoglycemic | 27/70 (27.8) | ref | 50/97 (57.2) | ref |
| Diabetic | 10/13 (43.5) | 1.96 (0.86–4.47) | 16/23 (69.6) | 2.63 (1.47–4.70) |
| *Overall blood glucose level, post-diagnostic, men* | | | | |
| Normoglycemic | 69/354 (19.5) | ref | 203/354 (57.3) | ref |
| Diabetic | 28/114 (24.6) | 1.55 (1.00–2.40) | 73/114 (64.0) | 1.40 (1.08–1.82) |
| *Overall blood glucose level, post-diagnostic, women* | | | | |
| Normoglycemic | 35/127 (27.6) | ref | 72/127 (56.7) | ref |
| Diabetic | 11/29 (37.9) | 1.63 (0.85–3.12) | 23/29 (79.3) | 1.78 (1.14–2.77) |
| *Overall blood glucose level, post-diagnostic, localized cancer* | | | | |
| Normoglycemic | 50/320 (19.5) | ref | 163/320 (50.9) | ref |
| Diabetic | 7/80 (8.8) | 0.85 (0.42–1.74) | 44/80 (55.0) | 1.50 (1.09–2.06) |
| *Overall blood glucose level, post-diagnostic, metastatic cancer* | | | | |
| Normoglycemic | 37/62 (59.7) | ref | 49/62 (79.0) | ref |
| Diabetic | 21/30 (70.0) | 2.89 (1.33–3.93) | 24/30 (80.0) | 1.84 (1.12–3.00) |
| *Overall blood glucose level, post-diagnostic, diagnose age <70 years* | | | | |
| Normoglycemic | 69/289 (23.9) | ref | 206/289 (71.3) | ref |
| Diabetic | 27/90 (30.0) | 2.04 (1.05–3.95) | 72/90 (64.0) | 1.52 (0.96–2.42) |
| *Overall blood glucose level, post-diagnostic, diagnose age >70 years* | | | | |
| Normoglycemic | 35/192 (18.2) | ref | 69/192 (35.9) | ref |
| Diabetic | 12/53 (22.6) | 1.57 (1.02–2.43) | 24/53 (45.3) | 1.56 (1.21–2.03) |

## Sensitivity analyses

We calculated competing risks regression analyses with non-cancer death as the competing risk. Results were similar as in the main analysis.

The association between post-diagnostic glucose level and cancer-specific or overall mortality was no longer significant in sensitivity analysis limited to cases with available information on tumor extent at diagnosis, HR 0.97 (95% CI 0.67–1.41) and HR 1.06, (95% CI 0.82–1.37), respectively.

## Discussion

In our population-based cohort study we found that BCa patients with diabetic glucose level had worse survival compared to normoglycemic ones as hypothesized, regardless of whether diabetic hyperglycemia occurred before or after BCa. Expectedly, also overall mortality was higher in diabetics compared to non-diabetics.

Hyperglycemia may be stronger independent predictor of BCa death than anti-DM medication use, as demonstrated by analyses including both variables: only hyperglycemia was an independent risk predictor, retaining statistically significant risk association. This indirectly supports hyperglycemia as supporting cancer progression, whereas antidiabetic medication use may mitigate the risk increase. However, both predicted overall mortality equally, suggesting that diabetes in general is a risk factor, presumably due to cardiovascular effects.

Based on subgroup analysis by gender, diabetics had a similar risk of BCa death both among men and women. Concordantly, two cohort studies of 2,110,074 participants observed similar difference in risk of BCa related death by diabetes

among men and women. Both of these studies had larger numbers of women compared to our study population. Proportion of fatal BCa among women was definitely higher in our study compared to the previous studies (proportion of BCa deaths in women with diabetes 28.9% vs. 5.2% and 7.3%, respectively,) [7,14]. This could suggest that our study cohort has more high-risk cases, allowing more statistical power to evaluate BCa-specific survival than in previous studies.

Anti-DM medication users underwent cystectomy less often than non-users. However, no difference in cystectomies was observed by glycemic status. A meta-analysis of 5 cohort studies showed that diabetes is significantly associated with the poor overall and cancer specific survival after radical cystectomy in bladder cancer patients [15]. The study also suggested that diabetes is a risk factor for life-threatening complications in cystectomy patients after surgery and is positively associated with serious complications. It is well-known that diabetics often have comorbidities affecting suitability for surgical treatment. Therefore, they might be treated in less aggressive manner [16]. Nevertheless, around 24% of participants with diabetic glycemic status underwent this operation. Uncontrolled diabetes has also been linked to increased risk of postoperative complications after cystectomy [17]. Oh et al. found elevated risk of BCa recurrence, cancer-specific and overall mortality among diabetics in a cohort of BCa patients who underwent cystectomy [18]. Increased risk of BCa death by diabetes was observed in our analysis after adjustment for primary BCa treatment. Therefore, the observed survival association is not explained by differing treatment alone.

One of the strengths of this study was large study population covering all newly diagnosed BCa cases in Finland during 1995–2012. Large study population reduces the effect of random error. We used four nationwide registers which enabled us to assess study population in a comprehensive manner. The use of prescription register linked to HILMO database complimented the information of chronic diseases such as diabetes and reduced the bias of possibly poorly recorded subsidiary diagnosis. Using data from nationwide registries eliminated biases often attributed to survey studies, such as recall bias. Additionally, detailed information on timing of blood glucose and HbA1c measurements allowed us to evaluate separately diabetes before and after BCa diagnosis. Therefore, we were in a position to evaluate the association between diabetes and BCa survival more comprehensively than before.

In our study cohort, the proportion of men was 76%, which is the usual gender ratio among BCa patients [3]. Nevertheless, the novelty of our study is that we managed to show the increased risk of BCa death concerning diabetic blood glucose levels in both genders. For our knowledge only one study has previously reported increased risk of BCa-specific death among women with diabetes [19]. Additionally, cohort study of 37,459 women showed that postmenopausal women with self-reported diabetes had 2.46-fold increased risk of BCa than postmenopausal women without diabetes [20]. This supports our notion that the risk is elevated among diabetics in both genders.

There were also limitations to this study. Blood glucose measurements were available to a subgroup only. Information on tobacco smoking was insufficient. To decrease this limitation, we adjusted the analysis for records of COPD and smoking from the HILMO database, but the information especially on smoking was severely limited. FCR registers only tumor extent as local vs. metastasized, whereas no information on muscle invasion or carcinoma in situ was available. Further, information on tumor extent on diagnosis was available only for 74% of the cohort. Limiting the analysis to cases with this information available abolished the risk association between post-diagnostic glucose level and mortality. This suggests that cases with missing information may cause bias in our results. Our results require confirmation from datasets with more comprehensive information on stage, possibly via use of multiple imputations.

Furthermore, we did not have information on intravesical treatments. We neither had information on diet, physical activity, occupation (chemical exposure, mostly in occupational exposure is known risk factor for BCa) or socioeconomic status that may have served as confounding factors. We were not able to separate diabetes for type 1 and 2 in our analysis due to low number of type 1 diabetics.

## Conclusions

Diabetic hyperglycemia is associated with increased BCa-specific mortality. Our study suggests diabetic glycemic status is an independent prognostic risk factor, whereas antidiabetic medication use may mitigate the risk increase to small degree.

## Author contributions

**Conceptualization:** Teuvo LJ Tammela.

**Data curation:** Teemu J. Murtola.

**Formal analysis:** Lauri Vuoristo, Silja M Vuorlaakso.

**Supervision:** Antti Pöyhönen, Andres Kotsar, Teuvo LJ Tammela, Teemu J. Murtola.

**Validation:** Teemu J. Murtola.

**Visualization:** Antti Pöyhönen.

**Writing – original draft:** Lauri Vuoristo, Silja M Vuorlaakso, Andres Kotsar.

**Writing – review & editing:** Antti Pöyhönen, Teuvo LJ Tammela, Teemu J. Murtola.

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
