## [Decision Letter · Decision Letter 0]

7 Jun 2024

Dear Dr. Murtola,

Thank you for submitting your manuscript to PLOS ONE. After careful consideration, we feel that it has merit but does not fully meet PLOS ONE’s publication criteria as it currently stands. Therefore, we invite you to submit a revised version of the manuscript that addresses the points raised during the review process.

We look forward to receiving your revised manuscript.

Kind regards,

Chen Li, Ph.D.

Academic Editor

PLOS ONE

Journal Requirements:

"Teuvo LJ Tammela: consultant fees from Astellas, Bayer and Roche.

Teemu J Murtola: consultant fees from Novartis, Astellas, Janssen Cilag, Amgen and Recordati. Lecture fees from Ferring, Novartis, Sanofi, Bayer, Roche, Pfizer, Ipsen, Astellas, Amgen and Janssen Cilag. Research funding from Bayer.

Other authors declare no conflicts of interest." 

Reviewers' comments:

Reviewer's Responses to Questions

**Comments to the Author**

1. Is the manuscript technically sound, and do the data support the conclusions?

Reviewer #1: Partly

Reviewer #2: Partly

Reviewer #3: Yes

Reviewer #4: Partly

2. Has the statistical analysis been performed appropriately and rigorously?

Reviewer #1: Yes

Reviewer #2: Yes

Reviewer #3: Yes

Reviewer #4: No

3. Have the authors made all data underlying the findings in their manuscript fully available?

Reviewer #1: Yes

Reviewer #2: Yes

Reviewer #3: Yes

Reviewer #4: No

4. Is the manuscript presented in an intelligible fashion and written in standard English?

Reviewer #1: Yes

Reviewer #2: Yes

Reviewer #3: No

Reviewer #4: Yes

Reviewer #1: The authors investigate the potential association between diabetes and BCa prognosis, but there are several questions that need to be answered before considering the publication.

1) The use of nationwide registries is commendable, but the specific criteria for inclusion and exclusion of data need to be detailed.

2) The limitations section is brief; a more thorough discussion of potential biases and their impact on the results is needed.

3) The study mentions the lack of smoking data but does not discuss how this might affect the results.

4) The study should discuss whether there are significant differences in the results between men and women.

5) More detailed information on drug resistance, including specific mechanisms, should be provided.

6) The generalisability of the results to other populations should be discussed.

7) The manuscript should highlight the novel aspects of this study compared to previous research.

8) The clinical implications of the study findings should be discussed in detail.

9) The limitations of the study, such as potential sources of bias and the generalisability of the results, should be clearly stated.

10) The manuscript should be carefully checked for grammatical errors, spelling mistakes, or unclear presentation.

Reviewer #2: In the submitted article, "Diabetes and Bladder Cancer Prognosis in a Finnish Population-based Cohort," Vuoristo et al. investigate the association between diabetes and bladder cancer prognosis in a Finnish cohort. They find that diabetic hyperglycemia post-diagnosis is linked to increased mortality, although the analysis is limited by insufficient smoking data and incomplete information on glycemic control. However, the study has several other limitations, with comments, suggestions, and concerns that need addressing, and the text in the manuscript requires significant modifications.

Main Concerns:

1. Variable Availability of Comorbidity Data: While diabetes and other comorbidities such as hypertension and obesity were accounted for, the methods section does not specify how consistently these were recorded across the study period. Variations in recording these data can affect the study's ability to accurately assess the impact of these comorbid conditions on bladder cancer outcomes.

2. Suboptimal Control of Confounders: The study utilized multivariable Cox regression models which adjusted for several factors. However, it's unclear if all potential confounders were adequately controlled. For instance, treatment modalities and tumor characteristics could also influence survival outcomes but were not extensively detailed in the context of their influence on the diabetes-cancer link. The author should include detailed analyses of how different treatments and tumor characteristics influence the diabetes-bladder cancer mortality link to provide deeper insights. This would help in understanding whether the observed associations are directly due to diabetes or mediated by other factors.

3. Statistical Significance in Subgroups: Some of the subgroup analyses, such as those involving pre-diagnostic glycemic levels, did not reach statistical significance. This raises questions about the robustness of these associations and whether they could be due to chance. The author should employ advanced statistical techniques that can handle missing data effectively, such as multiple imputation, which might mitigate some of the biases arising from incomplete data. Additionally, using stratified models or propensity score matching could better control for unmeasured confounders.

Text Problems:

1. Original: "Diabetic post-diagnostic blood glucose level was associated with worse BCa-specific HR 1.58 (1.10–2.28) and overall survival HR 1.50 (1.20–1.89)."

Suggestion: "The post-diagnostic blood glucose levels in diabetics were associated with a worse BCa-specific hazard ratio (HR) of 1.58 (1.10–2.28) and an overall survival HR of 1.50 (1.20–1.89)."

2. Original: "The risk increase was slightly lower among hyperglycemic DM medication users (HR 1.43 (0,68–3.02) than non-users HR 1.80 (1.13–2.85)."

Suggestion: "The increase in risk was slightly lower among hyperglycemic DM medication users, with an HR of 1.43 (0.68–3.02), compared to non-users, who had an HR of 1.80 (1.13–2.85)."

3. Original: "We assessed the risk of comorbidity by using Charlson comorbidity index (CCI)."

Suggestion: "We assessed the risk of comorbidity using the Charlson Comorbidity Index (CCI)."

4. Original: "Information was collected on fasting glucose and glycated haemoglobins (HbA1c) levels during 1995-2012 for a subset of study population living in the Pirkanmaa region in Finland."

Suggestion: "Information was collected on fasting glucose and glycated hemoglobin (HbA1c) levels from 1995-2012 for a subset of the study population living in the Pirkanmaa region of Finland."

5. Original: "Diagnoses made in 1995 were recorded as ICD-9 codes: 250 participants had diagnosis for diabetes, 272 for hypercholesterolemia, 401 for hypertension and 278 for obesity."

Suggestion: "Diagnoses made in 1995 were recorded using ICD-9 codes: 250 participants were diagnosed with diabetes, 272 with hypercholesterolemia, 401 with hypertension, and 278 with obesity."

Reviewer #3: The study includes a large cohort of 14,638 BCa patients diagnosed in Finland between 1995 and 2012. Utilizing data from three national databases, the study evaluates the impact of diabetes, measured through blood glucose levels and HbA1c, on BCa-specific and overall mortality using multivariable Cox regression models. The findings indicate that diabetic blood glucose levels are associated with worse BCa-specific and overall survival, with the increased risk being slightly mitigated among those using antidiabetic medication. The study highlights the need for further research to understand the mechanisms behind these associations and the potential implications for clinical management of BCa patients with diabetes.

Here are my questions based on the article:

1. How did the researchers ensure the completeness and accuracy of the data obtained from the three national databases, and what measures were taken to address any potential data gaps or inconsistencies?

2. What criteria were used to classify participants' glycemic status, and how did the study account for variations in blood glucose levels and HbA1c measurements over time?

3. How did the use of antidiabetic medication influence the survival outcomes of BCa patients, and what potential mechanisms might explain the observed differences in mortality risk between medication users and non-users?

4. What confounding factors were considered in the multivariable Cox regression models, and how did the study address potential confounders such as smoking and comorbidities?

5. How did the associations between diabetic glycemic status and BCa-specific mortality differ by gender and cancer stage, and what implications do these findings have for personalized treatment strategies in BCa patients with diabetes?

Here are the grammatical and phrasing errors found in the provided document along with corrections:

1. Location: Abstract (Lines 21-22)

- Original: "Diabetes might increase a risk of bladder cancer (BCa) incidence and bladder cancer specific death."

- Correction: "Diabetes might increase the risk of bladder cancer (BCa) incidence and bladder cancer-specific death."

2. Location: Abstract (Lines 27-28)

- Original: "Three databases were used to obtain information on BCa, diabetes and comorbidities."

- Correction: "Three databases were used to obtain information on BCa, diabetes, and comorbidities."

3. Location: Abstract (Lines 30-31)

- Original: "During median follow-up of 4.3 years after BCa diagnosis 3,582 (24.5 %) participants died of BCa."

- Correction: "During a median follow-up of 4.3 years after BCa diagnosis, 3,582 (24.5%) participants died of BCa."

4. Location: Abstract (Line 33)

- Original: "The risk increase was slightly lower among hyperglycemic DM medication users (HR 1.43 (0,68–3.02) than non-users HR 1.80 (1.13–2.85)."

- Correction: "The risk increase was slightly lower among hyperglycemic DM medication users (HR 1.43 (0.68–3.02)) than non-users (HR 1.80 (1.13–2.85))."

5. Location: Abstract (Line 37)

- Original: "This was observed in both genders."

- Correction: "This was observed in both men and women."

6. Location: Introduction (Line 43)

- Original: "Bladder cancer (BCa) is the ninth most common cancer worldwide [1]. Well-known risk factors are smoking, male gender, living in developed countries and chemical exposure [1,2]."

- Correction: "Bladder cancer (BCa) is the ninth most common cancer worldwide [1]. Well-known risk factors include smoking, being male, living in developed countries, and chemical exposure [1,2]."

7. Location: Introduction (Line 50)

- Original: "These presumably support growth of cancer cells by offering energy and growth factors for rapid cell growth."

- Correction: "These presumably support the growth of cancer cells by providing energy and growth factors for rapid cell growth."

8. Location: Results (Line 156)

- Original: "No clear differences in Charlson co-morbidity score or prevalence of COPD were observed by medication use or blood glucose level."

- Correction: "No clear differences in Charlson comorbidity score or prevalence of COPD were observed by medication use or blood glucose level."

9. Location: Results (Line 160)

- Original: "DM medication users had similar overall mortality than non-users, while hyperglycemic participants had higher overall mortality than normoglycemic participants."

- Correction: "DM medication users had similar overall mortality to non-users, while hyperglycemic participants had higher overall mortality than normoglycemic participants."

10. Location: Results (Line 174)

- Original: "The risk increase was slightly lower among hyperglycemic DM medication users HR 1.43 (0,68–3.02) than non-users HR 1.80 (1.13–2.85) but remained elevated compared to normoglycemia in both groups."

- Correction: "The risk increase was slightly lower among hyperglycemic DM medication users (HR 1.43 (0.68–3.02)) than non-users (HR 1.80 (1.13–2.85)) but remained elevated compared to normoglycemia in both groups."

11. Location: Discussion (Line 219)

- Original: "Expectedly, also overall mortality was higher in diabetics compared to non-diabetics."

- Correction: "As expected, overall mortality was also higher in diabetics compared to non-diabetics."

12. Location: Discussion (Line 240)

- Original: "Therefore, they might be treated in less aggressive manner."

- Correction: "Therefore, they might be treated in a less aggressive manner."

13. Location: Discussion (Line 258)

- Original: "For our knowledge only one study has previously reported increased risk of BCa-specific death among women with diabetes."

- Correction: "To our knowledge, only one study has previously reported an increased risk of BCa-specific death among women with diabetes."

14. Location: Discussion (Line 265)

- Original: "To decrease this limitation, we adjusted the analysis for records of COPD and smoking from the HILMO database, but the information especially on smoking was severely limited."

- Correction: "To mitigate this limitation, we adjusted the analysis for records of COPD and smoking from the HILMO database, but the information on smoking was particularly limited."

15. Location: Conclusions (Line 281)

- Original: "whereas antidiabetic medication use may mitigate the risk increase to small degree."

- Correction: "whereas antidiabetic medication use may mitigate the risk increase to a small degree."

Reviewer #4: Although this study is a large-scale investigation, encompassing almost all bladder cancer cases in Finland from 1995 to 2012, and explores the relationship between diabetes and bladder cancer prognosis through the analysis of blood glucose levels and antidiabetic medication history, it has two major limitations that significantly impact its clinical relevance.

Firstly, the data is outdated. The study covers diabetic patients from 1995 to 2012, but many new antidiabetic drugs have been introduced since 2012, such as SGLT-2 inhibitors and GLP-1 receptor agonists. These newer medications can significantly affect patients' weight, blood glucose levels, and even urine glucose, potentially influencing the final risk assessment. Therefore, I recommend the authors use more recent data for their analysis.

Secondly, there is a conceptual confusion. Not using antidiabetic drugs does not equate to not having diabetes. In fact, many patients can control their blood glucose levels solely through diet and exercise without needing medication, but this does not change the fact that they have diabetes or that they may have hyperinsulinemia. Thus, relying solely on the use of antidiabetic drugs to classify individuals as diabetic or non-diabetic is clearly inappropriate. Additionally, categorizing groups based on blood glucose levels is also flawed. When patients manage their blood glucose to normal levels through medication, they are still diabetic, their insulin levels have not improved, and insulin resistance remains. They may still be using antidiabetic drugs.

Minor issues

The study does not differentiate between type 1 and type 2 diabetes. As the authors mentioned from Line49-51, elevated blood glucose and high insulin levels are risk factors, and the fundamental difference between type 1 and type 2 diabetes is insulin levels. If the two types cannot be distinguished, using blood glucose alone to assess the risk of diabetes for cancer is inappropriate. Although this was explained in the limitations section, I suggest focusing the analysis on only one type of diabetes.

Additionally, the authors did not assess the impact of smoking, which is a significant risk factor for bladder cancer. The proportion of smokers is high among diabetic patients, which affects the accuracy of the conclusions. This should be further explained in the limitations section.

The study concludes that antidiabetic medications may have a protective effect in reducing the risk of bladder cancer. However, since the specific medication regimens of the patients are unclear, this conclusion is not accurate. In particular, the potential risk of bladder cancer associated with pioglitazone has garnered attention from various organizations, including the FDA, in recent years.

Why was acarbose excluded?

There are some grammatical and spelling errors in the articles, such as

Line 53-55, I suggest using “According to a meta-analysis of 36 studies, diabetes was associated with an increased risk of bladder cancer (BCa) in men【6】. A prospective population-based cohort study also found an increased risk of BCa death, but only in men【7】. Conversely, another study with 1,000 patients found no association between diabetes (type 1 or 2) and BCa risk or mortality【8】.”

Line67-69, I suggest using “The most common tumor morphologies were transitional cell carcinoma(12,027 cases, 82.0%), papillary transitional cell carcinoma( 1,626 cases ,11.1%), and unspecified malignant neoplasm( 533 cases ,3.6%).

**Do you want your identity to be public for this peer review?** For information about this choice, including consent withdrawal, please see our Privacy Policy

Reviewer #1: **Yes: ** Xiaodong Zou

Reviewer #2: No

Reviewer #3: **Yes: ** Peng Wang

Reviewer #4: **Yes: ** YUAN LIU

---

## [Author Response · Author response to Decision Letter 1]

24 Jan 2025

Reviewers' comments:

Reviewer's Responses to Questions

Comments to the Author

Reviewer #1:

The authors investigate the potential association between diabetes and BCa prognosis, but there are several questions that need to be answered before considering the publication.

1) The use of nationwide registries is commendable, but the specific criteria for inclusion and exclusion of data need to be detailed.

A: We included all recorded, newly diagnosed bladder cancer cases in Finland during the study period. Only exclusions were cases diagnosed before the age of 20 years, and cases diagnosed after the date of death, i.e. autopsy-diagnosed cases. This has been clarified in Materials and Methods, study cohort, 1st paragraph:

“Data was obtained from the Finnish Cancer Registry (FCR) for all newly diagnosed BCa cases in Finland 1995-2012.”

“Ten cases diagnosed under the age of 20 were excluded. Additionally, 182 cases were excluded in which the BCa diagnosis was made at the time of death or thereafter. “

2) The limitations section is brief; a more thorough discussion of potential biases and their impact on the results is needed.

A: We have expanded our Discussion on study limitations:

“There were also limitations to this study. Blood glucose measurements were available only for a subgroup. Information on tobacco smoking is not routinely recorded by national registries, thus the information was insufficient. Smoking may have confounded the results, likely raising the observed BCa mortality among anti-DM medication users, if smoking was more common among them.”

“Furthermore, we did not have information on intravesical treatments, like BCG instillations. However, these are unlikely to differ by anti-DM medication use. We neither had information on diet, physical activity, occupation (chemical exposure, mostly in occupational exposure is known risk factor for BCa) or socioeconomic status that may have served as confounding factors, exaggerating the observed risk differences.”

“Even though using of national registries allowed forming a population-based cohort of BCa cases covering the entire of Finland, it also meant heterogenous data with incomplete information on key variables like smoking or tumor extent. Therefore, further studies are needed with more detailed information to validate our findings.”

3) The study mentions the lack of smoking data but does not discuss how this might affect the results.

A: We now clarify that in the Discussion:

“Information on tobacco smoking is not routinely recorded by national registries, thus the information was insufficient. Smoking may have confounded the results, likely raising the observed BCa mortality among anti-DM medication users, if smoking was more common among them.”

4) The study should discuss whether there are significant differences in the results between men and women.

A: We ran subgroup analyses to test for this, but no significant differences in the results between men and women were observed. We report this in the Results, subgroup analyses-section: “Age and gender did not modify the risk association between diabetic blood glucose level and bladder cancer specific and overall survival (p for interaction > 0.05).”

We also highlight this in the Discussion:

“Based on subgroup analysis, hyperglycemia was a similar risk factor of BCa death both among men and women, suggesting that BCa patients may benefit from hyperglycemia management regardless of their gender. Concordantly, two cohort studies of 2,110,074 participants observed similar difference for BCa mortality by diabetes both among men and women. These studies had larger numbers of women compared to our study population. However, proportion of women with fatal BCa was higher in our study compared to the previous studies: proportion of BCa deaths among women was 28.9% in our study cohort vs. 5.2% and 7.3% in the previous studies [7,18]. This could suggest that our study cohort has more high-risk cases, allowing more statistical power to evaluate BCa-specific survival than previous studies.”

5) More detailed information on drug resistance, including specific mechanisms, should be provided.

A: This is a very broad topic, as multiple mechanisms of resistance have been reported to each antidiabetic drug group. It is also mainly beyond the scope of this article, as we are not able to evaluate such mechanisms in our register-based data.

Nevertheless, this is likely one of the reasons why diabetic glucose levels were observed also among antidiabetic medication users. Therefore, we have added the following paragraph to Discussion, 3rd paragraph:

“Hyperglyemia was observed even among DM medication users, which highlights the challenges of diabetes treatment. Resistance to anti-DM drugs may arise via multiple mechanisms, with impaired insulin sensitivity of peripheral tissues and compromised insulin production in pancreatic beta-cells being the most important. Insensitivity to metformin and targeted drugs like GLP-1 receptor agonists and DPP-4 inhibitors may arise via non-canonical activation of their targets (Rena 2017, Nauck 2016).“

New references:

Rena G, Hardie DG, Pearson ER. The mechanisms of action of metformin. Diabetologia. 201760:1577-1585.

Nauck M. Incretin therapies: highlighting common features and differences in the modes of action of glucagon-like peptide-1 receptor agonists and dipeptidyl peptidase-4 inhibitors. Diabetes Obes Metab. 2016;18:203-16.

6) The generalisability of the results to other populations should be discussed.

A: The following text is added to the Discussion section, line 288: “The Finnish population is mostly Caucasian, so generalizability of our results to other ethnicities is uncertain.”

7) The manuscript should highlight the novel aspects of this study compared to previous research.

A: The following text is included to the Discussion: “The novelty of our study is simultaneous evaluation of glycemic status and antidiabetic medication use as prognostic factors of BCa at population level.”

8) The clinical implications of the study findings should be discussed in detail.

A: In our opinion, the most important clinical implication from our study is, that BCa patients could benefit from effective blood glucose management.

We have added the following text 1st paragraph of the Discussion:

“This suggests, that effective follow-up and management of hyperglycemia in BCa patients may provide survival benefits.”

And to the Conclusion:

“Effective management of hyperglycemia is likely beneficial in BCa patients.”

9) The limitations of the study, such as potential sources of bias and the generalisability of the results, should be clearly stated.

A: We have expanded our Discussion (8th and 9th paragraph) on possible confounding by missing information on smoking, limited information on tumor stage and BCa treatments and missing information on lifestyle factors. We now also provide estimates to which direction these limitations may have biased our results.

10) The manuscript should be carefully checked for grammatical errors, spelling mistakes, or unclear presentation.

A: We have performed a language check, and have improved the clarity of presentation to our best ability. We are happy to revise further, if some specific parts of the manuscript remain unclear.

Reviewer #2:

In the submitted article, "Diabetes and Bladder Cancer Prognosis in a Finnish Population-based Cohort," Vuoristo et al. investigate the association between diabetes and bladder cancer prognosis in a Finnish cohort. They find that diabetic hyperglycemia post-diagnosis is linked to increased mortality, although the analysis is limited by insufficient smoking data and incomplete information on glycemic control. However, the study has several other limitations, with comments, suggestions, and concerns that need addressing, and the text in the manuscript requires significant modifications.

Main Concerns:

1. Variable Availability of Comorbidity Data: While diabetes and other comorbidities such as hypertension and obesity were accounted for, the methods section does not specify how consistently these were recorded across the study period. Variations in recording these data can affect the study's ability to accurately assess the impact of these comorbid conditions on bladder cancer outcomes.

A: We have added a clarification of this issue to Methods, Information on diabetes and comorbidities, last paragraph:

“We assessed the risk of comorbidity by using the Charlson comorbidity index (CCI) [15]. Diagnoses for comorbidities were obtained from HILMO and supplemented with medication data from the SII prescription database. These two registers complement each other, as HILMO database records only the diagnoses made in tertiary health care units, thus underestimating the full prevalence of conditions that are mainly managed in primary care, such as hypertension. Prescription database, on the other hand, records all medication purchases regardless of where they have been prescribed, but does not record the indication for prescription.”

2. Suboptimal Control of Confounders: The study utilized multivariable Cox regression models which adjusted for several factors. However, it's unclear if all potential confounders were adequately controlled. For instance, treatment modalities and tumor characteristics could also influence survival outcomes but were not extensively detailed in the context of their influence on the diabetes-cancer link. The author should include detailed analyses of how different treatments and tumor characteristics influence the diabetes-bladder cancer mortality link to provide deeper insights. This would help in understanding whether the observed associations are directly due to diabetes or mediated by other factors.

A: We agree in full. Unfortunately, detailed information on tumor characteristics, such as tumor grade or presence of carcinoma in situ lesions, nor detailed information on treatment, such as BCG instillations were not available from population-wide registries used for this study. We used the limited data we had available in the analysis, but is is clear that verification is needed from other study cohorts with more extensive data on tumor characteristics and treatment.

We discuss these limitations in the Discussion:

“FCR registers only tumor extent as local vs. metastasized, whereas no information on muscle invasion or carcinoma in situ was available. Further, information on tumor extent on diagnosis was available only for 74% of the cohort. Limiting the analysis to cases with this information available abolished the risk association between post-diagnostic glucose level and mortality. This suggests that cases with missing information may have caused bias. Furthermore, we did not have information on intravesical treatments, like BCG instillations. However, these are unlikely to differ by anti-DM medication use….. Therefore, our results need to be confirmed in other cohorts with more detailed information on clinical risk factors and comorbidities.”

3. Statistical Significance in Subgroups: Some of the subgroup analyses, such as those involving pre-diagnostic glycemic levels, did not reach statistical significance. This raises questions about the robustness of these associations and whether they could be due to chance. The author should employ advanced statistical techniques that can handle missing data effectively, such as multiple imputation, which might mitigate some of the biases arising from incomplete data. Additionally, using stratified models or propensity score matching could better control for unmeasured confounders.

A: It is true, that our results in general are hypothesis generating and require confirmation from further studies. Multiple imputation methods are uncertain to remove bias, as the imputations are made based on the assumptions of the researchers. Propensity scores are also calculated based on available data, and their value over multivariate adjusted models in large datasets such as ours has been questioned (Hadley J, JNCI 2010). Therefore, the assumptions and calculations for advanced mathematical modeling would require an article of it’s own, which would compare the results obtained with the imputed dataset to those in the present analysis based on recorded data only. In this article we aim to evaluate the impact of missing data in sensitivity analyses limiting the cohort to participants with the data available, as the large size of the cohort allows that in most cases and comparing the results to those using the full dataset.

Hadley J, Yabroff KR, Barrett MJ, Penson DF, Saigal CS, Potosky AL. Comparative effectiveness of prostate cancer treatments: evaluating statistical adjustments for confounding in observational data. J Natl Cancer Inst. 2010 Dec 1;102(23):1780-93. doi: 10.1093/jnci/djq393. Epub 2010 Oct 13. PMID: 20944078; PMCID: PMC2994860.

Text Problems:

1. Original: "Diabetic post-diagnostic blood glucose level was associated with worse BCa-specific HR 1.58 (1.10–2.28) and overall survival HR 1.50 (1.20–1.89)."

Suggestion: "The post-diagnostic blood glucose levels in diabetics were associated with a worse BCa-specific hazard ratio (HR) of 1.58 (1.10–2.28) and an overall survival HR of 1.50 (1.20–1.89)."

A: As defined in the methods, we categorized participants to normoglycemic or hyperglycemic based on their median fasting blood glucose or HbA1c. Hence the expression “Diabetic post-diagnostic blood glucose level”. All of the participants with blood glucose measurements available could have been diabetics, even some of those who had normoglycemia due to good treatment balance. Therefore, we have clarified the sentence as follows:

“Hyperglycemia at diabetic level after BCa diagnosis was associated..”

2. Original: "The risk increase was slightly lower among hyperglycemic DM medication users (HR 1.43 (0,68–3.02) than non-users HR 1.80 (1.13–2.85)."

Suggestion: "The increase in risk was slightly lower among hyperglycemic DM medication users, with an HR of 1.43 (0.68–3.02), compared to non-users, who had an HR of 1.80 (1.13–2.85)."

A: Corrected as suggested.

3. Original: "We assessed the risk of comorbidity by using Charlson comorbidity index (CCI)."

Suggestion: "We assessed the risk of comorbidity using the Charlson Comorbidity Index (CCI)."

A: Corrected as suggested.

4. Original: "Information was collected on fasting glucose and glycated haemoglobins (HbA1c) levels during 1995-2012 for a subset of study population living in the Pirkanmaa region in Finland."

Suggestion: "Information was collected on fasting glucose and glycated hemoglobin (HbA1c) levels from 1995-2012 for a subset of the study population living in the Pirkanmaa region of Finland."

A: Corrected as suggested.

5. Original: "Diagnoses made in 1995 were recorded as ICD-9 codes: 250 participants had diagnosis for diabetes, 272 for hypercholesterolemia, 401 for hypertension and 278 for obesity."

Suggestion: "Diagnoses made in 1995 were recorded using ICD-9 codes: 250 participants were diagnosed with diabetes, 272 with hypercholesterolemia, 401 with hypertension, and 278 with obesity."

A: Corrected as suggested.

Reviewer #3:

The study includes a large cohort of 14,638 BCa patients diagnosed in Finland between 1995 and 2012. Utilizing data from three national databases, the study evaluates the impact of diabetes, measured through blood glucose levels and HbA1c, on BCa-specific and overall mortality using multivariable Cox regression models. The findings indicate that diabetic blood glucose levels are associated with worse BCa-specific and overall survival, with the increased risk being slightly mitigated among those using antidiabetic medication. The study highlights the need for further research to understand the mechanisms behind these associations and the potential implications for clinical management of BCa patients with diabetes.

Here are my questions based on the article:

1. How did the researchers ensure the completeness and accuracy of the data obtained from the three national databases, and what measures were taken to address any potential data gaps or inconsistencies?

A: A text was added to the end of the fir

---

## [Decision Letter · Decision Letter 1]

4 Feb 2025

Dear Dr. Murtola,

Thank you for submitting your manuscript to PLOS ONE. After careful consideration, we feel that it has merit but does not fully meet PLOS ONE’s publication criteria as it currently stands. Therefore, we invite you to submit a revised version of the manuscript that addresses the points raised during the review process.

We look forward to receiving your revised manuscript.

Kind regards,

Robert Jeenchen Chen, MD, MPH, ChFC®, EA, CLU

Academic Editor

PLOS ONE

Journal Requirements:

Reviewers' comments:

Reviewer's Responses to Questions

**Comments to the Author**

Reviewer #1: All comments have been addressed

Reviewer #3: All comments have been addressed

2. Is the manuscript technically sound, and do the data support the conclusions?

Reviewer #1: Yes

Reviewer #3: Yes

3. Has the statistical analysis been performed appropriately and rigorously?

Reviewer #1: Yes

Reviewer #3: Yes

4. Have the authors made all data underlying the findings in their manuscript fully available?

Reviewer #1: Yes

Reviewer #3: Yes

5. Is the manuscript presented in an intelligible fashion and written in standard English?

Reviewer #1: Yes

Reviewer #3: Yes

Reviewer #1: (No Response)

Reviewer #3: The authors have thoroughly addressed all of my concerns. I have no further questions and recommend proceeding with the acceptance process according to the journal's guidelines.

**Do you want your identity to be public for this peer review?** For information about this choice, including consent withdrawal, please see our Privacy Policy

Reviewer #1: **Yes: ** Xiaodong Zou

Reviewer #3: **Yes: ** peng wang

---

## [Author Response · Author response to Decision Letter 2]

15 Feb 2025

Dear Dr. Chen,

Thank You for your interest in our research. As both reviewers were satisfied with our previous responses and revisions, and we received no further comments, we made no further revisions and submit here the previously revised version of our manuscript. We hope it will be suitable for publication in PLOS ONE.

---

## [Decision Letter · Decision Letter 2]

24 Mar 2025

Dear Dr. Murtola,

Thank you for submitting your manuscript to PLOS ONE. After careful consideration, we feel that it has merit but does not fully meet PLOS ONE’s publication criteria as it currently stands. Therefore, we invite you to submit a revised version of the manuscript that addresses the points raised during the review process.

We look forward to receiving your revised manuscript.

Kind regards,

Robert Jeenchen Chen, MD, MPH, ChFC®, EA, CLU

Academic Editor

PLOS ONE

Journal Requirements:

Reviewers' comments:

Reviewer's Responses to Questions

**Comments to the Author**

Reviewer #3: All comments have been addressed

Reviewer #5: All comments have been addressed

2. Is the manuscript technically sound, and do the data support the conclusions?

Reviewer #3: Yes

Reviewer #5: (No Response)

3. Has the statistical analysis been performed appropriately and rigorously?

Reviewer #3: Yes

Reviewer #5: (No Response)

4. Have the authors made all data underlying the findings in their manuscript fully available?

Reviewer #3: Yes

Reviewer #5: (No Response)

5. Is the manuscript presented in an intelligible fashion and written in standard English?

Reviewer #3: Yes

Reviewer #5: (No Response)

Reviewer #3: The authors have thoroughly addressed all of my concerns. I have no further questions and recommend proceeding with the acceptance process according to the journal's guidelines.

Reviewer #5: In this study, the authors investigate the potential association between diabetes and bladder cancer prognosis.

I think the revisions made by the authors in line with the referee suggestions are sufficient. I have no additional questions about the study. However, while correcting the grammar of the article, I think some words were forgotten to be deleted in the 3rd paragraph of the introduction.

**Do you want your identity to be public for this peer review?** For information about this choice, including consent withdrawal, please see our Privacy Policy

Reviewer #3: **Yes: ** peng wang

Reviewer #5: No

---

## [Author Response · Author response to Decision Letter 3]

9 Apr 2025

PONE-D-24-16036R2

Hyperglycemia and bladder cancer prognosis in a Finnish population-based cohort

PLOS ONE

Dear editor,

Thank You for possibility to answer these questions. We have thoroughly considered our answers to reviewer´s remarks and submit answers here and the revised version has been made with and without mark changed as instructed. We hope that after these corrections, our manuscript could be worth of publishing in PLOS ONE.

Journal Requirements:

RESPONSE: The reference list has been reviewed. One reference was changed to more proper one and three references were deleted. The style of reference list was changed to match PLOS ONE style.

1. Antoni S, Ferlay J, Soerjomataram I, Znaor A, Jemal A, Bray F. Bladder Cancer Incidence and Mortality: A Global Overview and Recent Trends. Eur Urol. 2017;71(1): 96-108.

2. Cumberbatch MGK, Jubber I, Black PC, Esperto F, Figueroa JD, Kamat AM, et al. Epidemiology of Bladder Cancer: A Systematic Review and Contemporary Update of Risk Factors in 2018. Eur Urol. 2018;74(6): 784-795.

3. Dobruch J, Daneshmand S, Fisch M, Lotan Y, Noon AP, Resnick MJ, et al. Gender and Bladder Cancer: A Collaborative Review of Etiology, Biology, and Outcomes. Eur Urol. 2016;69(2): 300-310.

4. Larsson SC, Orsini N, Brismar K, Wolk Aet al. Diabetes mellitus and risk of bladder cancer: a meta-analysis. Diabetologia. 2006;49(12): 2819-2823.

5. Zhou XH, Qiao Q, Zethelius B, Pyörälä K, Söderberg S, Pajak A, et al. Diabetes, prediabetes and cancer mortality. Diabetologia. 2010;53(9): 1867-1876.

6. Zhu Z, Wang X, Shen Z, Lu Y, Zhong S, Xu C. et al. Risk of bladder cancer in patients with diabetes mellitus: an updated meta-analysis of 36 observational studies. BMC Cancer. 2013;13: 310. doi: 10.1186/1471-2407-13-310.

7. Coughlin SS, Calle EE, Teras LR, Petrelli J, Thun MJ. et al. Diabetes Mellitus as a Predictor of Cancer Mortality in a Large Cohort of US Adults. Am J Epidemiol. 2004;159(12): 1160-1167.

8. Goossens ME, Zeegers MP, Bazelier MT, De Bruin ML, Buntinx F, de Vries F. et al. Risk of bladder cancer in patients with diabetes: a retrospective cohort study. BMJ Open. 2015;5(6): e007470. doi: 10.1136/bmjopen-2014-007470.

9. Teppo L, Pukkala E, Lehtonen M. Data Quality and Quality Control of a Population-Based Cancer Registry: Experience in Finland. Acta Oncol. 1994;33(4): 365-369.

10. Finnish Cancer Registry. http://cancerregistry.fi/

11. Sund R. Quality of the Finnish Hospital Discharge Register: A systematic review. Scand J Public Health. 2012;40(6): 505-515.

12. Klaukka T. The Finnish database on drug utilisation. Norwegian Journal of Epidemiology. 2001;11(1):19-22.

13. Finnish National Health Care Insurance. https://www.kela.fi/web/en/national-health-insurance

14. The Anatomical Therapeutic Chemical (ATC) classification system. https://www.whocc.no/atc_ddd_index/

15. Charlson ME, Carrozzino D, Guidi J, Patierno C. Charlson Comorbidity Index: A Critical Review of Clinimetric Properties. Psychother Psychosom. 2022;91: 8-35. Peterson J, Paget S, Lachs M, et al. The risk of comorbidity. Ann Rheum Dis. 2012;71(5):635-637.

16. Campbell PT, Newton CC, Patel AV, Jacobs EJ, Gapstur SM. et al. Diabetes and Cause-Specific Mortality in a Prospective Cohort of One Million U.S. Adults. Diabetes Care. 2012;35(9): 1835-1844.

17. Dong Lv, Ying Xiang, Tao Song, Guang-Peng Z, Tai-Ming S. et al. Diabetes Is a Risk Factor for the Prognosis of Patients with Bladder Cancer: A Meta-Analysis. J Oncol. 2022; 1997507. doi: 10.1155/2022/1997507.

18. Van de Poll-Franse LV, Houterman S, Janssen-Heijnen MLG, Dercksen MW Coebergh JWW, Haak HR. et al. Less aggressive treatment and worse overall survival in cancer patients with diabetes: A large population based analysis. Int J Cancer. 2007;120(9): 1986-1992.

19. Faiena I, Dombrovskiy VY, Sultan RC, Salmasi AH, Singer EA, Weiss RE. et al. Effect of Uncontrolled Diabetes on Outcomes After Cystectomy in Patients With Bladder Cancer: A Population-Based Study. Clin Genitourin Cancer. 2016;14(5): e509-e514.

20. Oh JJ, Kang MY, Jo JK, Lee HM, Byun SS, Lee SE et al. Association between diabetes mellitus and oncological outcomes in bladder cancer patients undergoing radical cystectomy. Int J Urol. 2015;22(12): 1112-1117.

21. Tseng CH. Diabetes and risk of bladder cancer: a study using the National Health Insurance database in Taiwan. Diabetologia. 2011;54(8): 2009-2015.

22. Tripathi A, Folsom AR, Anderson KE. Risk factors for urinary bladder carcinoma in postmenopausal women. Cancer. 2002;95(11): 2316–2323.

Reviewers' comments:

Reviewer's Responses to Questions

Comments to the Author

1. If the authors have adequately addressed your comments raised in a previous round of review and you feel that this manuscript is now acceptable for publication, you may indicate that here to bypass the “Comments to the Author” section, enter your conflict of interest statement in the “Confidential to Editor” section, and submit your "Accept" recommendation.

Reviewer #3: All comments have been addressed

Reviewer #5: All comments have been addressed

RESPONSE: No actions needed

2. Is the manuscript technically sound, and do the data support the conclusions?

Reviewer #3: Yes

Reviewer #5: (No Response)

RESPONSE: No actions needed

3. Has the statistical analysis been performed appropriately and rigorously?

Reviewer #3: Yes

Reviewer #5: (No Response)

RESPONSE: No actions needed

4. Have the authors made all data underlying the findings in their manuscript fully available?

The PLOS Data policy<http://track.editorialmanager.com/CL0/http:%2F%2Fwww.plosone.org%2Fstatic%2Fpolicies.action%23sharing/1/010f0195c693853b-e9391df4-831f-4c09-bc3d-cab85d5402cd-000000/59fVLGzW1aiED_EG1p4o-5qRD0lEkysVgQBquo65xh8=203> requires authors to make all data underlying the findings described in their manuscript fully available without restriction, with rare exception (please refer to the Data Availability Statement in the manuscript PDF file). The data should be provided as part of the manuscript or its supporting information, or deposited to a public repository. For example, in addition to summary statistics, the data points behind means, medians and variance measures should be available. If there are restrictions on publicly sharing data—e.g. participant privacy or use of data from a third party—those must be specified.

Reviewer #3: Yes

Reviewer #5: (No Response)

RESPONSE: No actions needed

5. Is the manuscript presented in an intelligible fashion and written in standard English?

Reviewer #3: Yes

Reviewer #5: (No Response)

RESPONSE: No actions needed

6. Review Comments to the Author

Reviewer #3: The authors have thoroughly addressed all of my concerns. I have no further questions and recommend proceeding with the acceptance process according to the journal's guidelines.

RESPONSE: Thank You.

Reviewer #5: In this study, the authors investigate the potential association between diabetes and bladder cancer prognosis.

I think the revisions made by the authors in line with the referee suggestions are sufficient. I have no additional questions about the study. However, while correcting the grammar of the article, I think some words were forgotten to be deleted in the 3rd paragraph of the introduction.

RESPONSE: Thank You. The 3rd paragraph of the introduction has been checked and corrected.

7. PLOS authors have the option to publish the peer review history of their article (what does this mean?<https://track.editorialmanager.com/CL0/https:%2F%2Fjournals.plos.org%2Fplosone%2Fs%2Feditorial-and-peer-review-process%23loc-peer-review-history/1/010f0195c693853b-e9391df4-831f-4c09-bc3d-cab85d5402cd-000000/rZ0UH0RJsU11eawvF74c2CZpE-uit5dx4NNTYvRn_Ds=203>). If published, this will include your full peer review and any attached files.

Do you want your identity to be public for this peer review? For information about this choice, including consent withdrawal, please see our Privacy Policy<https://track.editorialmanager.com/CL0/https:%2F%2Fwww.plos.org%2Fprivacy-policy/1/010f0195c693853b-e9391df4-831f-4c09-bc3d-cab85d5402cd-000000/mQPv1YvGfC3vtt3IRaxLvp4w_K9jYXiU5k7nIJEg8mM=203>.

Reviewer #3: Yes: peng wang

Reviewer #5: No

---

## [Decision Letter · Decision Letter 3]

14 Apr 2025

Dear Dr. Murtola,

Thank you for submitting your manuscript to PLOS ONE. After careful consideration, we feel that it has merit but does not fully meet PLOS ONE’s publication criteria as it currently stands. Therefore, we invite you to submit a revised version of the manuscript that addresses the points raised during the review process.

We look forward to receiving your revised manuscript.

Kind regards,

Robert Jeenchen Chen, MD, MPH, ChFC®, EA, CLU

Academic Editor

PLOS ONE

Journal Requirements:

Reviewers' comments:

Reviewer's Responses to Questions

**Comments to the Author**

Reviewer #3: All comments have been addressed

Reviewer #5: All comments have been addressed

2. Is the manuscript technically sound, and do the data support the conclusions?

Reviewer #3: Yes

Reviewer #5: (No Response)

3. Has the statistical analysis been performed appropriately and rigorously?

Reviewer #3: Yes

Reviewer #5: (No Response)

4. Have the authors made all data underlying the findings in their manuscript fully available?

Reviewer #3: Yes

Reviewer #5: (No Response)

5. Is the manuscript presented in an intelligible fashion and written in standard English?

Reviewer #3: Yes

Reviewer #5: (No Response)

Reviewer #3: ALL COMMENTS HAVE BEEN ADDRESSED. The authors have thoroughly addressed all of my concerns. I have no further questions and recommend proceeding with the acceptance process according to the journal's guidelines.

Reviewer #5: (No Response)

**Do you want your identity to be public for this peer review?** For information about this choice, including consent withdrawal, please see our Privacy Policy

Reviewer #3: **Yes: ** peng wang

Reviewer #5: No

---

## [Author Response · Author response to Decision Letter 4]

16 Apr 2025

Journal Requirements:

RESPONSE:

The entire reference list was reviewed. One reference was changed to more proper one and three references were deleted. The style of reference list was changed to match PLOS ONE style.

1. Antoni S, Ferlay J, Soerjomataram I, Znaor A, Jemal A, Bray F. Bladder Cancer Incidence and Mortality: A Global Overview and Recent Trends. Eur Urol. 2017;71(1): 96-108.

2. Cumberbatch MGK, Jubber I, Black PC, Esperto F, Figueroa JD, Kamat AM, et al. Epidemiology of Bladder Cancer: A Systematic Review and Contemporary Update of Risk Factors in 2018. Eur Urol. 2018;74(6): 784-795.

3. Dobruch J, Daneshmand S, Fisch M, Lotan Y, Noon AP, Resnick MJ, et al. Gender and Bladder Cancer: A Collaborative Review of Etiology, Biology, and Outcomes. Eur Urol. 2016;69(2): 300-310.

4. Larsson SC, Orsini N, Brismar K, Wolk Aet al. Diabetes mellitus and risk of bladder cancer: a meta-analysis. Diabetologia. 2006;49(12): 2819-2823.

5. Zhou XH, Qiao Q, Zethelius B, Pyörälä K, Söderberg S, Pajak A, et al. Diabetes, prediabetes and cancer mortality. Diabetologia. 2010;53(9): 1867-1876.

6. Zhu Z, Wang X, Shen Z, Lu Y, Zhong S, Xu C. et al. Risk of bladder cancer in patients with diabetes mellitus: an updated meta-analysis of 36 observational studies. BMC Cancer. 2013;13: 310. doi: 10.1186/1471-2407-13-310.

7. Coughlin SS, Calle EE, Teras LR, Petrelli J, Thun MJ. et al. Diabetes Mellitus as a Predictor of Cancer Mortality in a Large Cohort of US Adults. Am J Epidemiol. 2004;159(12): 1160-1167.

8. Goossens ME, Zeegers MP, Bazelier MT, De Bruin ML, Buntinx F, de Vries F. et al. Risk of bladder cancer in patients with diabetes: a retrospective cohort study. BMJ Open. 2015;5(6): e007470. doi: 10.1136/bmjopen-2014-007470.

9. Teppo L, Pukkala E, Lehtonen M. Data Quality and Quality Control of a Population-Based Cancer Registry: Experience in Finland. Acta Oncol. 1994;33(4): 365-369.

10. Finnish Cancer Registry. http://cancerregistry.fi/

11. Sund R. Quality of the Finnish Hospital Discharge Register: A systematic review. Scand J Public Health. 2012;40(6): 505-515.

12. Klaukka T. The Finnish database on drug utilisation. Norwegian Journal of Epidemiology. 2001;11(1):19-22.

13. Finnish National Health Care Insurance. https://www.kela.fi/web/en/national-health-insurance

14. The Anatomical Therapeutic Chemical (ATC) classification system. https://www.whocc.no/atc_ddd_index/

15. Charlson ME, Carrozzino D, Guidi J, Patierno C. Charlson Comorbidity Index: A Critical Review of Clinimetric Properties. Psychother Psychosom. 2022;91: 8-35. Peterson J, Paget S, Lachs M, et al. The risk of comorbidity. Ann Rheum Dis. 2012;71(5):635-637.

16. Campbell PT, Newton CC, Patel AV, Jacobs EJ, Gapstur SM. et al. Diabetes and Cause-Specific Mortality in a Prospective Cohort of One Million U.S. Adults. Diabetes Care. 2012;35(9): 1835-1844.

17. Dong Lv, Ying Xiang, Tao Song, Guang-Peng Z, Tai-Ming S. et al. Diabetes Is a Risk Factor for the Prognosis of Patients with Bladder Cancer: A Meta-Analysis. J Oncol. 2022; 1997507. doi: 10.1155/2022/1997507.

18. Van de Poll-Franse LV, Houterman S, Janssen-Heijnen MLG, Dercksen MW Coebergh JWW, Haak HR. et al. Less aggressive treatment and worse overall survival in cancer patients with diabetes: A large population based analysis. Int J Cancer. 2007;120(9): 1986-1992.

19. Faiena I, Dombrovskiy VY, Sultan RC, Salmasi AH, Singer EA, Weiss RE. et al. Effect of Uncontrolled Diabetes on Outcomes After Cystectomy in Patients With Bladder Cancer: A Population-Based Study. Clin Genitourin Cancer. 2016;14(5): e509-e514.

20. Oh JJ, Kang MY, Jo JK, Lee HM, Byun SS, Lee SE et al. Association between diabetes mellitus and oncological outcomes in bladder cancer patients undergoing radical cystectomy. Int J Urol. 2015;22(12): 1112-1117.

21. Tseng CH. Diabetes and risk of bladder cancer: a study using the National Health Insurance database in Taiwan. Diabetologia. 2011;54(8): 2009-2015.

22. Tripathi A, Folsom AR, Anderson KE. Risk factors for urinary bladder carcinoma in postmenopausal women. Cancer. 2002;95(11): 2316–2323.

reference [11] was changed as it was not found in Pubmed

1. Klaukka T. The Finnish database on drug utilisation. Norwegian Journal of Epidemiology. 2001;11(1):19-22.

We replaced the reference with:

Jormanainen V, Relander T, Jormanainen V, Lindgren M. Decreasing Number of Medication Prescriptions After e-Prescriptions Became Mandatory and Their Valid Period Was Extended: A Big Bang Policy Change in Finland in 2017. Stud Health Technol Inform. 2020;270: 833-837.

Reviewers' comments:

Reviewer's Responses to Questions

Comments to the Author

1. If the authors have adequately addressed your comments raised in a previous round of review and you feel that this manuscript is now acceptable for publication, you may indicate that here to bypass the “Comments to the Author” section, enter your conflict of interest statement in the “Confidential to Editor” section, and submit your "Accept" recommendation.

Reviewer #3: All comments have been addressed

Reviewer #5: All comments have been addressed

RESPONSE: We are thankful for the positive remarks

2. Is the manuscript technically sound, and do the data support the conclusions?

Reviewer #3: Yes

Reviewer #5: (No Response)

RESPONSE: We are thankful for the positive remarks

3. Has the statistical analysis been performed appropriately and rigorously?

Reviewer #3: Yes

Reviewer #5: (No Response)

RESPONSE: We are thankful for the positive remarks

4. Have the authors made all data underlying the findings in their manuscript fully available?

The PLOS Data policy<http://track.editorialmanager.com/CL0/http:%2F%2Fwww.plosone.org%2Fstatic%2Fpolicies.action%23sharing/1/010f0195c693853b-e9391df4-831f-4c09-bc3d-cab85d5402cd-000000/59fVLGzW1aiED_EG1p4o-5qRD0lEkysVgQBquo65xh8=203> requires authors to make all data underlying the findings described in their manuscript fully available without restriction, with rare exception (please refer to the Data Availability Statement in the manuscript PDF file). The data should be provided as part of the manuscript or its supporting information, or deposited to a public repository. For example, in addition to summary statistics, the data points behind means, medians and variance measures should be available. If there are restrictions on publicly sharing data—e.g. participant privacy or use of data from a third party—those must be specified.

Reviewer #3: Yes

Reviewer #5: (No Response)

RESPONSE: We are thankful for the positive remarks

5. Is the manuscript presented in an intelligible fashion and written in standard English?

Reviewer #3: Yes

Reviewer #5: (No Response)

RESPONSE: We are thankful for the positive remarks

6. Review Comments to the Author

Reviewer #3: The authors have thoroughly addressed all of my concerns. I have no further questions and recommend proceeding with the acceptance process according to the journal's guidelines.

RESPONSE: Thank You.

Reviewer #5: In this study, the authors investigate the potential association between diabetes and bladder cancer prognosis.

I think the revisions made by the authors in line with the referee suggestions are sufficient. I have no additional questions about the study. However, while correcting the grammar of the article, I think some words were forgotten to be deleted in the 3rd paragraph of the introduction.

RESPONSE: Thank You. The 3rd paragraph of the introduction has been checked and corrected.

7. PLOS authors have the option to publish the peer review history of their article (what does this mean?<https://track.editorialmanager.com/CL0/https:%2F%2Fjournals.plos.org%2Fplosone%2Fs%2Feditorial-and-peer-review-process%23loc-peer-review-history/1/010f0195c693853b-e9391df4-831f-4c09-bc3d-cab85d5402cd-000000/rZ0UH0RJsU11eawvF74c2CZpE-uit5dx4NNTYvRn_Ds=203>). If published, this will include your full peer review and any attached files.

Do you want your identity to be public for this peer review? For information about this choice, including consent withdrawal, please see our Privacy Policy<https://track.editorialmanager.com/CL0/https:%2F%2Fwww.plos.org%2Fprivacy-policy/1/010f0195c693853b-e9391df4-831f-4c09-bc3d-cab85d5402cd-000000/mQPv1YvGfC3vtt3IRaxLvp4w_K9jYXiU5k7nIJEg8mM=203>.

Reviewer #3: Yes: peng wang

Reviewer #5: No

---

## [Decision Letter · Decision Letter 4]

22 Apr 2025

Dear Dr. Murtola,

Thank you for submitting your manuscript to PLOS ONE. After careful consideration, we feel that it has merit but does not fully meet PLOS ONE’s publication criteria as it currently stands. Therefore, we invite you to submit a revised version of the manuscript that addresses the points raised during the review process.

We look forward to receiving your revised manuscript.

Kind regards,

Robert Jeenchen Chen, MD, MPH, ChFC®, EA

Academic Editor

PLOS ONE

Journal Requirements:

Reviewers' comments:

Reviewer's Responses to Questions

**Comments to the Author**

Reviewer #3: All comments have been addressed

Reviewer #5: (No Response)

2. Is the manuscript technically sound, and do the data support the conclusions?

Reviewer #3: Yes

Reviewer #5: Yes

3. Has the statistical analysis been performed appropriately and rigorously?

Reviewer #3: Yes

Reviewer #5: I Don't Know

4. Have the authors made all data underlying the findings in their manuscript fully available?

Reviewer #3: Yes

Reviewer #5: No

5. Is the manuscript presented in an intelligible fashion and written in standard English?

Reviewer #3: Yes

Reviewer #5: Yes

Reviewer #3: The authors have thoroughly addressed all of my concerns. I have no further questions and recommend proceeding with the acceptance process according to the journal's guidelines.

Reviewer #5: I want to thank the authors for the revisions made to the article. (Location: Materials and methods, Lines 98-99 ) The authors removed the 14th reference (‘’14. The Anatomical Therapeutic Chemical (ATC) classification system. https://www.whocc.no/atc_ddd_index/ ‘’) from the article, but the referenced text was not removed, and no reference was provided. I recommend that this situation be explained, or that the appropriate reference be added.

**Do you want your identity to be public for this peer review?** For information about this choice, including consent withdrawal, please see our Privacy Policy

Reviewer #3: **Yes: ** peng wang

Reviewer #5: No

---

## [Author Response · Author response to Decision Letter 5]

1 Jun 2025

PONE-D-24-16036R2

Hyperglycemia and bladder cancer prognosis in a Finnish population-based cohort

PLOS ONE

Dear editor,

Thank You for possibility to answer these questions. We have thoroughly considered our answers to reviewer´s remarks and submit answers here and the revised version has been made with and without mark changed as instructed. We hope that after these corrections, our manuscript could be worth of publishing in PLOS ONE.

Journal Requirements:

RESPONSE: Reference list has been checked and one reference was added based on reviewer´s comment.

12. WHO Collaborating Centre for Drug Statistics Methodology,

Guidelines for ATC classification and DDD assignment 2024. Oslo, Norway, 2023. ISBN 978-82-8406-428-4

Reviewers' comments:

Reviewer's Responses to Questions

Comments to the Author

1. If the authors have adequately addressed your comments raised in a previous round of review and you feel that this manuscript is now acceptable for publication, you may indicate that here to bypass the “Comments to the Author” section, enter your conflict of interest statement in the “Confidential to Editor” section, and submit your "Accept" recommendation.

Reviewer #3: All comments have been addressed

Reviewer #5: (No Response)

6. Review Comments to the Author

Reviewer #3: The authors have thoroughly addressed all of my concerns. I have no further questions and recommend proceeding with the acceptance process according to the journal's guidelines.

RESPONSE: Thank you.

Reviewer #5: I want to thank the authors for the revisions made to the article. (Location: Materials and methods, Lines 98-99 ) The authors removed the 14th reference (‘’14. The Anatomical Therapeutic Chemical (ATC) classification system. https://www.whocc.no/atc_ddd_index/ ‘’) from the article, but the referenced text was not removed, and no reference was provided. I recommend that this situation be explained, or that the appropriate reference be added.

RESPONSE: This is the relevant comment and we have added the appropriate published reference for this text, instead of a website:

12 WHO Collaborating Centre for Drug Statistics Methodology,

Guidelines for ATC classification and DDD assignment 2024. Oslo, Norway, 2023. ISBN 978-82-8406-428-4

---

## [Decision Letter · Decision Letter 5]

5 Jun 2025

Dear Dr. Murtola,

Thank you for submitting your manuscript to PLOS ONE. After careful consideration, we feel that it has merit but does not fully meet PLOS ONE’s publication criteria as it currently stands. Therefore, we invite you to submit a revised version of the manuscript that addresses the points raised during the review process.

We look forward to receiving your revised manuscript.

Kind regards,

Robert Jeenchen Chen, MD, MPH, ChFC®, EA

Academic Editor

PLOS ONE

Journal Requirements:

Reviewers' comments:

Reviewer's Responses to Questions

**Comments to the Author**

Reviewer #3: All comments have been addressed

Reviewer #5: All comments have been addressed

2. Is the manuscript technically sound, and do the data support the conclusions?

Reviewer #3: Yes

Reviewer #5: Yes

3. Has the statistical analysis been performed appropriately and rigorously?

Reviewer #3: Yes

Reviewer #5: I Don't Know

4. Have the authors made all data underlying the findings in their manuscript fully available?

Reviewer #3: Yes

Reviewer #5: No

5. Is the manuscript presented in an intelligible fashion and written in standard English?

Reviewer #3: Yes

Reviewer #5: Yes

Reviewer #3: The authors have thoroughly addressed all of my concerns. I have no further questions and recommend proceeding with the acceptance process according to the journal's guidelines.

Reviewer #5: I want to thank the authors for their thorough consideration and revision of all my concerns. The reference numbers have been updated following the revisions to the article. I believe some of the previously changed reference numbers were not removed from the text, so I recommend checking. (Example: Materials and methods, Lines 100, ‘’We assessed the risk of comorbidity by using Charlson comorbidity index (CCI) [1213].’’) I have no additional questions or suggestions, and I recommend proceeding with the acceptance process according to the journal's guidelines.

**Do you want your identity to be public for this peer review?** For information about this choice, including consent withdrawal, please see our Privacy Policy

Reviewer #3: **Yes: ** peng wang

Reviewer #5: No

---

## [Author Response · Author response to Decision Letter 6]

9 Jul 2025

Review Comments to the Author

Reviewer #3: The authors have thoroughly addressed all of my concerns. I have no further questions and recommend proceeding with the acceptance process according to the journal's guidelines.

RESPONSE: Thank you for the helpful review.

Reviewer #5: I want to thank the authors for their thorough consideration and revision of all my concerns. The reference numbers have been updated following the revisions to the article. I believe some of the previously changed reference numbers were not removed from the text, so I recommend checking. (Example: Materials and methods, Lines 100, ‘’We assessed the risk of comorbidity by using Charlson comorbidity index (CCI) [1213].’’) I have no additional questions or suggestions, and I recommend proceeding with the acceptance process according to the journal's guidelines.

RESPONSE: Thank you for the helpful review. We have now checked the reference numbers and removed any redundant ones from the text.

---

## [Decision Letter · Decision Letter 6]

4 Aug 2025

Dear Dr. Murtola,

Thank you for submitting your manuscript to PLOS ONE. After careful consideration, we feel that it has merit but does not fully meet PLOS ONE’s publication criteria as it currently stands. Therefore, we invite you to submit a revised version of the manuscript that addresses the points raised during the review process.

We look forward to receiving your revised manuscript.

Kind regards,

Robert Jeenchen Chen, MD, MPH, ChFC®, EA

Academic Editor

PLOS ONE

Journal Requirements:

Reviewers' comments:

Reviewer's Responses to Questions

**Comments to the Author**

Reviewer #5: All comments have been addressed

Reviewer #6: (No Response)

2. Is the manuscript technically sound, and do the data support the conclusions?

Reviewer #5: Yes

Reviewer #6: Yes

3. Has the statistical analysis been performed appropriately and rigorously?

Reviewer #5: I Don't Know

Reviewer #6: Yes

4. Have the authors made all data underlying the findings in their manuscript fully available?

Reviewer #5: No

Reviewer #6: No

5. Is the manuscript presented in an intelligible fashion and written in standard English?

Reviewer #5: Yes

Reviewer #6: Yes

Reviewer #5: I want to thank the authors for their thorough consideration and revision of all my concerns. I have no additional questions or suggestions, and I recommend proceeding with the acceptance process according to the journal's guidelines.

Reviewer #6: The manuscript is technically sound. The research question is clear and well-defined, the study design (a large, population-based cohort) is appropriate, and the conclusions are well-supported by the data. The statistical methodology, including the use of multivariable Cox regression and time-varying covariates, is appropriate for the research question and data structure.

Strengths:

Large, nationally representative cohort (n=14,638).

Linkage of multiple reliable registries enhances data validity.

Sensitivity and subgroup analyses (including competing risks) strengthen the robustness of findings.

Thoughtful discussion of limitations and contextualization within existing literature.

The statistical analysis is largely rigorous and appropriate. The use of multivariable Cox regression and time-dependent covariates is commendable. The authors correctly adjusted for relevant covariates and provided person-time denominators when reporting mortality rates.

Missing data on tumor stage and primary treatment (available for only 74% and 80%, respectively) were not imputed. Although a complete-case sensitivity analysis was conducted, use of multiple imputation would strengthen the conclusions.

Dichotomization of continuous glycemic variables (e.g., glucose, HbA1c) may reduce statistical power. Consider exploring dose-response relationships or using categories aligned with clinical thresholds in future work.

I recommend the manuscript be accepted after minor revisions or clarifications, particularly regarding the impact of missing data and the potential use of multiple imputation for future studies.

**Do you want your identity to be public for this peer review?** For information about this choice, including consent withdrawal, please see our Privacy Policy

Reviewer #5: No

Reviewer #6: No

---

## [Author Response · Author response to Decision Letter 7]

18 Sep 2025

PONE-D-24-16036R2

Hyperglycemia and bladder cancer prognosis in a Finnish population-based cohort

PLOS ONE

Dear editor,

Thank You for possibility to answer these questions. We have thoroughly considered our answers to reviewer´s remarks and submit answers here and the revised version has been made with and without mark changed as instructed. We hope that after these corrections, our manuscript could be worth of publishing in PLOS ONE.

Journal Requirements:

RESPONSE: We did not receive such recommendations

RESPONSE: Reference list has been checked

Reviewers' comments:

Reviewer's Responses to Questions

Comments to the Author

1. If the authors have adequately addressed your comments raised in a previous round of review and you feel that this manuscript is now acceptable for publication, you may indicate that here to bypass the “Comments to the Author” section, enter your conflict of interest statement in the “Confidential to Editor” section, and submit your "Accept" recommendation.

Reviewer #5: All comments have been addressed

Reviewer #6: (No Response)

2. Is the manuscript technically sound, and do the data support the conclusions?

Reviewer #5: Yes

Reviewer #6: Yes

3. Has the statistical analysis been performed appropriately and rigorously?

Reviewer #5: I Don't Know

Reviewer #6: Yes

4. Have the authors made all data underlying the findings in their manuscript fully available?

Reviewer #5: No

Reviewer #6: No

5. Is the manuscript presented in an intelligible fashion and written in standard English?

Reviewer #5: Yes

Reviewer #6: Yes

6. Review Comments to the Author

Reviewer #5: I want to thank the authors for their thorough consideration and revision of all my concerns. I have no additional questions or suggestions, and I recommend proceeding with the acceptance process according to the journal's guidelines.

Reviewer #6: The manuscript is technically sound. The research question is clear and well-defined, the study design (a large, population-based cohort) is appropriate, and the conclusions are well-supported by the data. The statistical methodology, including the use of multivariable Cox regression and time-varying covariates, is appropriate for the research question and data structure.

Strengths:

Large, nationally representative cohort (n=14,638).

Linkage of multiple reliable registries enhances data validity.

Sensitivity and subgroup analyses (including competing risks) strengthen the robustness of findings.

Thoughtful discussion of limitations and contextualization within existing literature.

The statistical analysis is largely rigorous and appropriate. The use of multivariable Cox regression and time-dependent covariates is commendable. The authors correctly adjusted for relevant covariates and provided person-time denominators when reporting mortality rates.

Missing data on tumor stage and primary treatment (available for only 74% and 80%, respectively) were not imputed. Although a complete-case sensitivity analysis was conducted, use of multiple imputation would strengthen the conclusions.

RESPONSE: We see the point. In our opinion, however, data imputation for such large portion of the data, 20-26%, also risks introducing bias based on the assumptions of the imputation. This is the reason we rather did a complete-case analysis. For do mention the limited information on stage as study limitation in the Discussion. We also give recommendation for further studies:

The FCR registers only tumor extent as local vs. metastasized, whereas no information on muscle invasion or carcinoma in situ was available. Further, information on tumor extent on diagnosis was available only for 74% of the cohort. Limiting the analysis to cases with this information available abolished the risk association between post-diagnostic glucose level and mortality. This suggests that cases with missing information may cause bias in our results. Our results require confirmation from datasets with more comprehensive information on stage, possibly via use of multiple imputations.

Dichotomization of continuous glycemic variables (e.g., glucose, HbA1c) may reduce statistical power. Consider exploring dose-response relationships or using categories aligned with clinical thresholds in future work.

RESPONSE: The categories for glycemic variables were indeed stratified by clinical thresholds: 7 mmol/l for fasting plasma glucose (international definition for diabetes mellitus is fasting plasma glucose ≥ 7,0 mmol/l. For HbA1c 53 mmol/mol was the threshold for diabetes at the time of the study period, although it has been lowered since.

We now clarify this in the text, lines 107-110:

“Men were defined as normoglycaemic, or diabetic separately for each follow-up years based on local and international thresholds (<53 to ≥53 mmol/mol, and ≤7 to >7 mmol/l) for yearly mean HbA1c or median fasting glucose, respectively”

I recommend the manuscript be accepted after minor revisions or clarifications, particularly regarding the impact of missing data and the potential use of multiple imputation for future studies.

RESPONSE: Thank you for your constructive comments.

7. PLOS authors have the option to publish the peer review history of their article (what does this mean?). If published, this will include your full peer review and any attached files.

Do you want your identity to be public for this peer review? For information about this choice, including consent withdrawal, please see our Privacy Policy.

Reviewer #5: No

Reviewer #6: No

---

## [Decision Letter · Decision Letter 7]

20 Nov 2025

Hyperglycemia and bladder cancer prognosis in a Finnish population-based cohort

PONE-D-24-16036R7

Dear Dr. Murtola

We’re pleased to inform you that your manuscript has been judged scientifically suitable for publication and will be formally accepted for publication once it meets all outstanding technical requirements.

Kind regards,

Filomena de Nigris, Ph.D.

Academic Editor

PLOS ONE

Additional Editor Comments (optional):

Reviewers' comments:

Reviewer's Responses to Questions

**Comments to the Author**

Reviewer #5: All comments have been addressed

Reviewer #7: All comments have been addressed

2. Is the manuscript technically sound, and do the data support the conclusions?

Reviewer #5: Yes

Reviewer #7: Yes

3. Has the statistical analysis been performed appropriately and rigorously?

Reviewer #5: I Don't Know

Reviewer #7: Yes

4. Have the authors made all data underlying the findings in their manuscript fully available?

Reviewer #5: No

Reviewer #7: Yes

5. Is the manuscript presented in an intelligible fashion and written in standard English?

Reviewer #5: Yes

Reviewer #7: Yes

Reviewer #5: I want to thank the authors for their thorough review and revisions. I have no further questions or suggestions, and I recommend moving forward with the acceptance process following the journal's guidelines.

Reviewer #7: Dear Authors,

Thank you for the thorough revisions and the clear and comprehensive responses provided.The changes you have made satisfactorily address all the points raised and further enhance the clarity and methodological rigor of the manuscript.

We have no additional comments and confirm the recommendation to proceed with acceptance of the manuscript in accordance with the journal’s guidelines.

**Do you want your identity to be public for this peer review?** For information about this choice, including consent withdrawal, please see our Privacy Policy

Reviewer #5: No

Reviewer #7: No

---

## [Editor Report · Acceptance letter]

PONE-D-24-16036R7

PLOS ONE

Dear Dr. Murtola,

I'm pleased to inform you that your manuscript has been deemed suitable for publication in PLOS ONE. Congratulations! Your manuscript is now being handed over to our production team.

Kind regards,

on behalf of

Prof. Filomena de Nigris

Academic Editor

PLOS ONE